# Obesity Is Associated with Changes in Laboratory Biomarkers in Chilean Patients Hospitalized with COVID-19

**DOI:** 10.3390/jcm12103392

**Published:** 2023-05-10

**Authors:** Sharon Viscardi, Luis Marileo, Hugo Delgado, Andrés San Martín, Loreto Hernández, Paola Garcés, Dina Guzmán-Oyarzo, Rodrigo Boguen, Gustavo Medina, Pablo Letelier, Ines Villano, Neftalí Guzmán

**Affiliations:** 1Laboratorio de Investigación en Salud de Precisión, Departamento de Procesos Diagnósticos y Evaluación, Facultad de Ciencias de la Salud, Universidad Católica de Temuco, Manuel Montt 56, Campus San Francisco, Temuco 4813302, Chile; sviscardi@uct.cl (S.V.);; 2Núcleo de Investigación en Producción Alimentaria, Universidad Católica de Temuco, Rudecindo Ortega 02950, Temuco 4813302, Chile; 3Biotechnology of Functional Foods Laboratory, Camino Sanquilco, Parcela 18, La Araucanía, Padre Las Casas 4850827, Chile; 4Programa de Doctorado en Ciencias Agropecuarias, Facultad de Recursos Naturales, Universidad Católica de Temuco, Rudecindo Ortega 02950, La Araucanía, Temuco 4813302, Chile; 5Clinical Laboratory, Hospital Dr. Hernán Henríquez Aravena, Temuco 4813302, Chile; 6Complejo Asistencial Padre Las Casas, Padre Las Casas 4850827, Chile; 7RedSalud Mayor, Temuco 4813302, Chile; 8Escuela de Tecnología Médica, Facultad de Medicina y Ciencias, Universidad San Sebastián Sede Concepción, Concepción 4080871, Chile; 9Department of Experimental Medicine, Section of Human Physiology and Unit of Dietetics and Sports Medicine, Università degli Studi della Campania “Luigi Vanvitelli”, 80100 Naples, Italy

**Keywords:** COVID-19, SARS-CoV-2, obesity, laboratory markers

## Abstract

Background and aims: It is reported that patients with obesity are more frequently hospitalized for COVID-19, and evidence exists that obesity is a risk factor, regardless of other comorbidities. The objective of this study was to evaluate the association of obesity with changes in laboratory biomarkers in hospitalized Chilean patients. Materials and methods: A total of 202 hospitalized patients (71 with obesity and 131 without obesity) with a diagnosis of COVID-19 were included in the study. Demographic, clinical, and laboratory (days 1, 3, 7, 15) data were obtained. We performed a statistical analysis, assuming significance with a value of *p* < 0.05. Results: Significant differences in chronic respiratory pathology are observed between patients with and without obesity. The inflammatory markers CPR, ferritin, NLR, and PLR are elevated during the evaluated period, while changes in leukocyte populations are present on day 1 (eosinophils) and day 3 (lymphocytes). Finally, a persistent elevation of D-dimer level is observed, presenting significant differences on day 7 between patients with and without obesity. Obesity had a positive correlation with admission to the critical patient unit, invasive mechanical ventilation, and length of hospital stay. Conclusion: Patients with obesity hospitalized for COVID-19 present marked elevations of inflammatory and hemostasis parameters, with a correlation between obesity, changes in laboratory biomarkers, and the risk of adverse clinical outcomes also observed.

## 1. Introduction

The COVID-19 pandemic has been a hazard to global public health since March 2020, when the International Virus Classification Commission titled the novel betacoronavirus-induced pneumonia coronavirus disease 2019 (COVID-19), a severe acute respiratory syndrome coronavirus 2 (SARS-CoV-2) [1]. As of January 2023, 670.191.206 laboratory confirmed human cases of COVID-19 have been notified to the WHO, with 6.823.120 deaths, meaning that the disease spectrum and transmission efficiency of this virus differs from those of SARS-CoV [2,3,4,5]. The virus spread aggressively and quickly, infecting millions of people worldwide, and generating high mortality and morbidity [6]. In diseased patients, SARS-CoV-2 mostly infects the type II pneumocytes in lungs, but also infects proximal tubule epithelial cells in the kidney in severe cases; such cells express its receptor angiotensin-converting enzyme 2 (ACE2), facilitating viral entry [7,8,9]. Infection leads to the downregulation of ACE2, which impacts the function of angiotensin II (Ang II) and renin–angiotensin system (RAS) in a variety of tissues, including lung, heart, vasculature, and kidney, and may enable the progression of COVID-19 from mild and moderate to more severe disease [6]. Among the more severe cases of COVID-19 requiring hospitalization, alterations such as lymphocytopenia, along with shortness of breath that may advance to acute respiratory distress syndrome (ARDS) requiring mechanical ventilation, are frequently observed [10,11,12]. Studies concentrating on the clinical features of infection show that patients with underlying comorbidities are more likely to progress to severe respiratory disease and even death. Among these, aging, arterial hypertension, diabetes, and obesity (body mass index (BMI) > 30 kg/m^2^) seem to confer the highest risk for the development of severe cases [6], although the majority of individuals eventually recover from infection, even those needing long hospitalization [10,12,13]. In particular, patients with obesity are at augmented risk of exacerbations from viral respiratory infections, leading to mechanical ventilation and potential non-survival [14]. This has important implications on global health as excess weight, usually represented by a raised BMI, affects vast numbers of people worldwide: 39% of adults are overweight (BMI ≥ 25.0 to 29.9 kg/m^2^) and 13% have clinical obesity (BMI ≥ 30.0 kg/m^2^) globally [15]. This is due to the decrease in total respiratory system compliance associated with obesity conditions, but also because in adipose tissue, the expression of ACE2 is higher than in lung tissue, the main target tissue affected by COVID-19 and other viruses like influenza [14,16,17]. Another relevant factor is that the obesity condition is associated with an excess of adipose tissue, which can gradually cause and/or aggravate a diversity of comorbidities, as well as hypertension, type 2 diabetes mellitus, dyslipidemia, and cardiovascular diseases [18]. Moreover, the obese state is linked with either increased or decreased total lymphocytes in the peripheral blood [18,19]. In this context, it is important to note that obesity is characterized by adipose tissue renovation, and pro-inflammatory alteration of the adipokine profile [20]. During pandemics, individuals with obesity are classified as clinically vulnerable groups, especially those with severe obesity (BMI > 40 kg/m^2^) [21]. For this reason, the objective of the study was to evaluate the association of obesity with changes in laboratory biomarkers in Chilean patients hospitalized for COVID-19.

## 2. Materials and Methods

### 2.1. Study Design and Participants

Through a retrospective study, a total of 202 hospitalized patients (>18 years old) of whom 71 had obesity (BMI ≥ 30 kg/m^2^) and 131 did not present obesity, with a diagnosis of COVID-19, and who were treated at the Hospital Dr. Hernán Henríquez Aravena, Temuco, Chile, were included. Exclusion criteria considered children under 18 years of age, pregnant women, and low-weight patients. All the patients were diagnosed according to established criteria (WHO interim guidance) by a reverse transcriptase real-time polymerase chain reaction (RT-PCR) assay of nasal and pharyngeal swab specimens, between March 2020 and April 2021. The severity of COVID-19 was defined according to the WHO Clinical Progression Scale [22] in moderate disease and severe disease.

The study was approved by the Scientific Ethics Committee of the Araucanía Sur Health Service (No. 144/2020) and was carried out in accordance with the ethical requirements established in the Declaration of Helsinki.

### 2.2. Data Collection

For the group of subjects included in the study, epidemiological, demographic, and clinical data were obtained from the medical history of each patient. Demographic variables, signs, and symptoms were recorded. In addition, laboratory biomarker results were obtained from the laboratory information system (LIS), at days 1, 3, 7, and 15 of hospitalization. All the samples for day 1 were collected within the first 24 h of hospital admission. The results were obtained by complete and differential blood count of leukocyte population through hematology analyzer MINDRAY CAL 6000 (Mindray, China), obtaining the neutrophil–lymphocyte ratio (NLR) and platelet–lymphocyte ratio (PLR). Hemostasis tests (prothrombin time, activated partial thromboplastin time, and D-dimer were performed by clotting time (PT and APTT) and immunoturbidimetric (D-dimer) assays, using a STA^®^ R Max coagulation analyzer (Stago, Asnières sur Seine, France). Measurements of inflammatory biomarkers were performed using the cobas 8000 series modular analyzer (Roche Diagnostics), by immunoturbidimetric assay (PCR) and electrochemiluminescence (ferritin).

### 2.3. Statistical Analysis

Chi-square was used to determine significant differences (*p* < 0.05) between the population with and without obesity that presented some comorbidity or clinical manifestation, whilst Student’s t-test was used to evaluate differences (*p* < 0.05) over time (day 1, 3, 7, and 15) of the variations of laboratory biomarkers. The relationship between obesity and laboratory parameters was evaluated over time (day 1, 3, 7, and 15) using Pearson’s correlation, and categorical principal component analysis (CATPCA) was used to determine a pattern over time (day 1, 3, 7, and 15) of the patients with obesity with respect to the evaluated laboratory parameters. Statistical analyses were performed in the IBM SPSS Statistics for Windows program (Version 26.0. IBM Corp., Armonk, NY, USA).

## 3. Results

### 3.1. Comorbidities and Clinical Manifestations of Chilean Patients with Obesity Hospitalized with COVID-19

A comparison was made between the population with or without obesity in terms of comorbidities and clinical manifestations (Table 1). On evaluating comorbidities, there is a significant difference for chronic respiratory pathologies, where a greater number of patients with obesity present chronic respiratory disease (8.4% of patients), compared to 4.5% of patients without obesity (*p* = 0.001).

In clinical manifestations, patients without obesity represent a segment with a greater number of cases reported for coughing (37.1%, *p* = 0.041) and dyspnea (37.1%, *p* = 0.002), whereas for clinical manifestations such as anosmia and dysgeusia there is a greater predominance in patients with obesity (3.5%, *p* = 0.042 for both cases). In addition, no significant differences were observed regarding symptoms before hospitalization between the groups (5.43 + 3.8 days in patients with obesity vs. 4.73 + 5.1 days in patients without obesity (*p* = 0.162).

### 3.2. Dynamic Variations of Laboratory Biomarkers in Chilean Patients with COVID-19

The existence of dynamic changes of the laboratory parameters between patients with and without obesity was determined (Table 2). When evaluating hematological parameters, an elevation of NLR and PLR was observed in patients hospitalized for moderate and severe COVID-19, with no significant differences between the groups. In addition, eosinopenia in the first 7 days of hospitalization was observed in all patients, regardless of whether they had or not obesity. At day 3, patients with obesity presented a higher lymphocyte count than those without this condition (1.2 and 1.0 respectively).

Regarding inflammation biomarkers, a significant increase in CRP and ferritin during the first 15 days of hospitalization was observed in all patients included in the study. On day 7, the patients without obesity had significantly higher levels of CRP than patients with obesity (61.1 and 51.7 mg/L, respectively). On the other hand, no significant differences were observed in PT and APTT between the groups; however, a sustained elevation of D-dimer was evidenced. On day 7, patients with obesity had higher D-dimer values (7.7 μg/mL) than those without obesity (2.3 μg/mL).

Figure 1 shows the dynamics of the patterns of the laboratory parameters. In relation to day 1, it is observed that patients with obesity are distributed in quadrants II and IV. Quadrant II presents a direct association in dimension 1 with the WBC (r = 0.675), neutrophils (r = 0.774), and NLR (r = 0.859), whereas quadrant IV is linked to dimension 2 with prothrombin (r = 0.057) and lymphocytes (r = 0.503). As for day 3, patients with obesity are distributed in quadrant I, with the main direct association being the WBC (r = 0.675, dim 1), neutrophils (r = 0.774, dim 1), and MCH (r = 0.286, dim 2) parameters. In quadrant II, hematocrit (r = 0.553, dim 2) and lymphocytes (r = 0.503, dim 2) are directly associated, and NLR (r = 0.859, dim 1) and PLR (r = 0.627, dim 1) are directly associated with quadrant III. In addition, regarding day 7, patients with obesity show the following distributions: in quadrant I, a direct association with WBC (r = 0.674, dim 2) and neutrophils (r = 0.718, dim 1); in quadrant II, a direct association with RBC (r = 0.519, dim 2), hematocrit (r = 0.493, dim 2); and PLT (r = 0.348, dim 2); and in quadrant III, a direct association with PLR (r = 0.370, dim 1). Finally, day 15 shows that patients with obesity show a direct association with lymphocytes (r = 0.597, dim 2), RBC (r = 0.466, dim 1), and PLT (r = 0.631, dim 1) in quadrant I; and with monocytes (r = 0.683, dim 1), WBC (r = 0.670, dim 1), neutrophils (r = 0.517, dim 1), and APTT1 (r = 0.282, dim 1) in quadrant IV.

### 3.3. Association between Laboratory Parameters and Obesity of Chilean Patients Hospitalized with COVID-19

Pearson’s statistical analysis was used to determine significant correlations between the laboratory parameters and obesity in patients (Table 3), highlighting the time variations of these parameters (day 1, 3, 7, and 15). For day 1, there is a significant direct correlation between WBC (r = 0.146, *p* = 0.038) and patients with obesity; likewise, a significant direct correlation is found between patients with obesity for the neutrophil parameter (r = 0.151, *p* = 0.032). By day 3, there is a statistically significant direct correlation between patients with obesity and WBC (r = 0.155, *p* = 0.042). On day 7, a significant direct correlation persists between the patients with obesity and WBC (r = 0.199, *p* = 0.016), as well as a significant direct correlation with NLR (r = 0.167, *p* = 0.045) and neutrophils (r = 0.219, *p* = 0.008). On the other hand, there is a statistically significant inverse correlation with patients with obesity for the MCHC parameter at this time point (r = –0.168, *p* = 0.043). By day 15, no significant direct or inverse correlations between the laboratory tests and the patients (with and without obesity) are observed.

### 3.4. Clinical Outcomes

Table 4 presents the association of obesity with the length of hospital stay (days), intensive care unit (ICU) admission, invasive mechanical ventilation (IMV), and death. No significant differences were observed regarding death; in contrast, significant differences were observed in admission to the critical patient unit, invasive mechanical ventilation, and length of hospital stay.

## 4. Discussion

Since obesity constitutes a recognized risk factor for severity and mortality in individuals infected with SARS-CoV-2, in this retrospective study, we discuss the association of obesity with changes in laboratory biomarkers in Chilean patients hospitalized for COVID-19.

It has been reported that patients with obesity are more frequently hospitalized for SARS-CoV-2 [23]. Additionally, obesity is a risk factor, regardless of other comorbidities, for the development of severe conditions and mortality in patients with COVID-19 [8]. Our results highlight that 8.4% of patients with obesity present chronic respiratory disease, and 4.5% of patients without obesity present this pathology (*p* = 0.001). The results show that patients without obesity have a high frequency of type 2 diabetes mellitus and cardiovascular disease. These results could be explained, at least in part, by the characteristics of the population included in the study, with an important component of the Native American population (Mapuche people) and high multidimensional poverty. Various studies show that the Mapuche population presents a greater susceptibility to diabetes and cardiometabolic risk factors, especially in those who migrate from rural to urban areas. A study shows that conditions such as urbanization and sedentary behavior influence insulin resistance to a greater extent in Chilean Mapuches than in Chileans of European descent [24]. In addition, a recent study shows that lifestyle differences between Native Americans (Mapuche and Aymara) and non-ethnic Chilean peers > 15 years are significantly associated with differences in blood pressure and glycaemia [25].

When we studied the clinical manifestations, patients without obesity represented the segment with the highest number of cases reported for coughing (25.2%, *p* = 0.041) and dyspnea (37.1%, *p* = 0.002), while for symptoms such as anosmia and dysgeusia there was a greater incidence in patients with obesity (3.5%, *p* = 0.042 for both cases). To our knowledge, such differences in clinical manifestations within the general population, with or without obesity, have not been described before [26].

Moreover, in this work we analyzed dynamic changes in laboratory biomarkers, comparing the population of patients with obesity and those without this condition (Table 2). In both groups included in the study, a significant increase in NLR and PLR was observed, which reaffirms the usefulness of these systemic inflammation indicator algorithms as prognostic indicators in patients with COVID-19. A previous study has shown that elevated NLR is associated with a greater risk of severity [27].

Regarding the leukocyte populations, the presence of eosinopenia was observed during the first 7 days of hospitalization. Curiously, Cortés-Vieyra et al. [28] reported that the population with obesity presented higher levels of eosinophils on day 1. These cells have a beneficial function in COVID-19 patients, probably by contributing to the control of the exacerbated inflammation induced by neutrophils, which can influence the outcome of the disease [28].

In addition, a significant increase in CRP and ferritin was observed during the first 15 days of hospitalization in both groups of patients included in the study. Various authors have proposed the importance of the marked elevation of CRP and hyperferritinemia as predictors of severity and mortality [29,30,31]. Recent evidence shows that the elevation of inflammation markers could be related to the presence of co-infection or secondary infection in patients with COVID-19. Thus, the results show that 4% of the patients included in the study (4.2% of patients with obesity and 3.8% of patients without obesity) presented secondary infection during hospitalization, a significantly lower frequency than that described in a recent meta-analysis [32]. Additionally, Mason et al. (2021) describe some laboratory biomarkers that would allow the exclusion of the presence of bacterial co-infection in patients with COVID-19 [33].

Moreover, our study also found a positive correlation between hospitalized patients with obesity and the parameters of WBC (r = 0.146, *p* = 0.038) and neutrophils (r = 0.151, *p* = 0.032) (Table 3). This association was maintained on days 1, 3, and 7. Few studies have reported the levels of immunoglobulins or complement in COVID-19 patients with or without obesity. Our results suggest that patients with obesity have higher levels of lymphocytes on day 3, while in patients without obesity the findings reported by Bobcakova et al. showed lower levels [34]. On the other hand, it should be noticed that the D-dimer concentration provides important information on the prognosis of COVID-19 patients, and in our study, patients with obesity presented higher D-dimer values on day 7. This should be considered an important risk factor, since it has been reported that a decrease in D-dimer levels may correlate with a clinical improvement in patients, and the increase in the value of D-dimer is the most sensitive change in the coagulation parameters in COVID-19, indicating a greater risk for the development of thrombosis [35].

In this context, Lu and Wang [36] described that the condition of patients worsened in the first week after admission, as shown by decreases in neutrophils, lymphocytes, monocytes, eosinophils, RBC, hemoglobin, NLR, PLT, and PLR. On the 7th day of admission, the levels of these latter markers decreased to their lowest values, although the RDW and CRP levels remained elevated. In our study, there is a statistically significant inverse correlation with patients with obesity for the MCHC parameter (r = –0.168, *p* = 0.043). Mao et al. [37] associated a decrease in MCHC with lung involvement, oxygen demand, and disease activity. Finally, in contrast with Lu and Wang (32), on day 15 the laboratory parameters analyzed did not correlate significantly with the patients evaluated in this study.

When evaluating the association of obesity with clinical results, no significant differences were observed regarding death; however, significant differences were observed in admission to the critical patient unit, invasive mechanical ventilation, and length of hospital stay (27.32 + 19.09 days in patients with obesity vs. 22.56 + 22.62 days without obesity, *p* = 0.011). In addition, obesity was observed to be a risk factor for intensive care unit (ICU) admission (odds ratio = 2.636; IC 1.433–4.848; *p* = 0.0016) and invasive mechanical ventilation (IMV) (odds ratio = 2.964; IC 1.631–5.388; *p* = 0.0003).A meta-analysis considering 12,591 patients showed that obesity was associated with an increased need for ICU intervention and invasive mechanical ventilation [38]. In addition, a prospective cohort study that evaluated the association between obesity and COVID-19 in 6.9 million people demonstrated a linear increase in the risk of severe disease, as well as of admission to critical care units [39].

Our study has some limitations. First, a limited number of patients were included, which corresponds to the casuistry of a single public clinical center in southern Chile. Second, the patients’ medical history may have some incomplete records due to the fact that in clinical practice subjective global assessment is performed.

## 5. Conclusions

The results allow us to conclude that patients with obesity hospitalized for COVID-19 present marked elevation of some inflammatory and hemostasis parameters, with a correlation between obesity and changes in laboratory biomarkers also observed. The most frequent previous health conditions were arterial hypertension, cardiovascular diseases, and diabetes; however, significant differences were observed in chronic respiratory disease between patients with and without obesity. The inflammatory markers CPR, ferritin, NLR, and PLR were elevated during the evaluated period, while changes in leukocyte populations were present on day 1 (eosinophils) and day 3 (lymphocytes). Finally, a persistent elevation of D-dimer was observed, presenting significant differences on day 7 between patients with and without obesity. Obesity was associated with admission to the critical patient unit, invasive mechanical ventilation, and length of hospital stay. This study contributes to the knowledge of risk stratification on the prognosis of COVID-19 hospitalized patients with obesity.

## Figures and Tables

**Figure 1 jcm-12-03392-f001:**
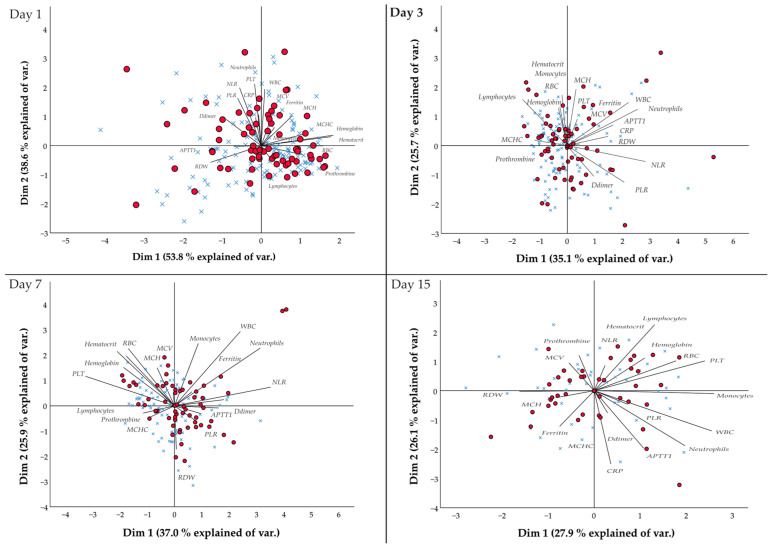
Principal component analysis (PCA) of patients with obesity with respect to the evaluated laboratory parameters.

**Table 1 jcm-12-03392-t001:** Demographic and clinical data of Chilean patients hospitalized with COVID-19.

	Total		With Obesity		Without Obesity		
	n = 202 (%)		n	(%)	n	(%)	*p*-Value
**Demographics**							
* Male*	95	(47.0)	28	(13.9)	67	(33.2)	0.111
* Female*	107	(53.0)	43	(21.3)	64	(31.7)	
* Age*	60.8 ± 14.8		58.9 ± 1.6		61.8 ± 1.3		0.483
**Comorbidity**							
* Arterial hypertension*	116	(57.4)	45	(22.3)	71	(35.1)	0.208
* Mellitus diabetes 2 (DM2)*	70	(34.7)	27	(13.4)	43	(21.3)	0.458
* Cardiovascular disease*	118	(58.4)	45	(22.3)	73	(36.1)	0.292
* Chronic respiratory pathology*	26	(12.9)	17	(8.4)	9	(4.5)	**0.001 ***
* Chronic kidney disease*	22	(10.9)	10	(5.0)	12	(5.9)	0.283
* Chronic liver disease*	8	4.0	2	(1.0)	6	(3.0)	0.540
**Clinical manifestations**							
* Fever*	115	(56.9)	45	(22.3)	70	(34.7)	0.173
* Odynophagia*	33	(16.3)	16	(7.9)	17	(8.4)	0.079
* Gastrointestinal symptoms*	43	(21.3)	15	(7.4)	28	(13.9)	0.967
* Cough*	126	(62.4)	51	(25.2)	75	(37.1)	**0.041 ***
* Headache*	53	(26.2)	21	(10.4)	32	(15.8)	0.427
* Dyspnea*	131	(64.9)	56	(27.7)	75	(37.1)	**0.002 ***
* Myalgia*	68	(33.7)	23	(11.4)	45	(22.3)	0.779
* Fatigue*	36	(17.8)	12	(5.9)	24	(11.9)	0.801
* Anosmia*	11	(5.4)	7	(3.5)	4	(2.0)	**0.042 ***
* Dysgeusia*	11	(5.4)	7	(3.5)	4	(2.0)	**0.042 ***

(*****) in bold represents significant differences (*p* < 0.05) using chi-square test between the population with obesity and without obesity.

**Table 2 jcm-12-03392-t002:** Dynamic variations of laboratory biomarkers in Chilean patients with COVID-19.

Parameter	Reference	Day 1						Day 3						Day 7						Day 15					
		With Obesity			Without Obesity			With Obesity			Without Obesity			With Obesity			Without Obesity			With Obesity			Without Obesity		
*Hematocrit, %*	35.0–47.0	37.2	±	0.8	36.4	±	0.6	36.0	±	0.7	35.2	±	0.6	34.7	±	0.7	34.7	±	0.8	32.0	±	0.8	31.0	±	0.9
*Hemoglobin, gr/dL*	14.0–17.5	12.6	±	0.3	12.3	±	0.2	12.0	±	0.3	11.8	±	0.2	11.4	±	0.3	11.6	±	0.3	10.5	±	0.3	10.2	±	0.3
*RBC, × 10^6^ µL*	3.80–5.80	4.3	±	0.1	4.2	±	0.1	4.0	±	0.1	4.0	±	0.1	4.0	±	0.1	3.9	±	0.1	3.5	±	0.1	3.4	±	0.1
*WBC, × 10^3^ µL*	4.00–12.00	8.7	±	0.4	7.7	±	0.3	9.4	±	0.6	8.0	±	0.4	10.4	±	0.7	8.6	±	0.4	9.9	±	0.5	9.2	±	0.6
*PLT, × 10^3^ µL*	150–450	232.2	±	10.2	219.3	±	10.0	262.8	±	13.3	261.0	±	12.7	301.5	±	14.2	299.3	±	15.3	267.2	±	20.4	287.1	±	22.6
*MCV, fL*	82.0–95.0	88.1	±	0.7	88.7	±	0.5	88.9	±	0.7	89.3	±	0.5	89.9	±	0.8	89.8	±	0.5	90.9	±	0.9	91.2	±	0.8
*MCH, pg*	25.0–32.0	29.6	±	0.3	29.9	±	0.2	29.5	±	0.2	29.8	±	0.2	29.1	±	0.4	29.8	±	0.2	29.6	±	0.4	29.9	±	0.3
*MCHC, gr/dL*	32.0–36.0	33.5	±	0.2	33.7	±	0.1	33.2	±	0.1	33.4	±	0.1	32.7	±	0.2	33.2	±	0.1	32.6	±	0.2	32.8	±	0.2
*RDW, %*	11.0–16.0	14.0	±	0.2	13.8	±	0.1	13.9	±	0.2	14.1	±	0.2	14.2	±	0.2	14.8	±	0.5	14.8	±	0.3	15.6	±	0.8
*Lymphocytes, ×10^3^ µL*	0.84–4.2	1.1	±	0.1	1.2	±	0.1	**1.2**	**±**	**0.2 ***	**1.0**	**±**	**0.1 ***	1.0	±	0.1	1.2	±	0.1	1.1	±	0.1	1.2	±	0.1
*Monocytes, ×10^3^ µL*	0.16–0.96	0.6	±	0.1	0.5	±	0.0	0.6	±	0.0	0.5	±	0.0	0.7	±	0.1	0.6	±	0.0	0.6	±	0.0	0.7	±	0.0
*Neutrophils, ×10^3^ µL*	2–8.2	6.9	±	0.4	5.9	±	0.3	7.6	±	0.6	6.4	±	0.4	8.5	±	0.7	6.6	±	0.4	7.9	±	0.5	7.2	±	0.6
*Eosinophils, × 10^3^ µL*	0.08–0.6	**0.03**	**±**	**0.0 ***	**0.01**	**±**	**0.0 ***	0.1	±	0.0	0.1	±	0.0	0.1	±	0.0	0.1	±	0.0	0.2	±	0.2	0.1	±	0.2
*Basophils, ×10^3^ µL*	0–0.12	0.0	±	0.0	0.0	±	0.0	0.0	±	0.0	0.1	±	0.1	0.0	±	0.0	0.0	±	0.0	0.0	±	0.0	0.0	±	0.0
*NLR*	0.107–3.19	8.6	±	0.9	7.4	±	0.7	11.7	±	1.6	9.6	±	1.3	12.3	±	1.6	8.4	±	1.2	10.9	±	2.0	8.0	±	0.9
*PLR*	46.79–218.01	277.7	±	26.8	250.1	±	16.4	355.8	±	34.9	329.8	±	21.6	385.3	±	35.5	310.5	±	23.9	326.8	±	54.5	305.0	±	31.8
*CRP (μg/L)*	<5	125.2	±	16.6	102.7	±	8.2	86.3	±	11.3	95.1	±	9.6	**51.7**	**±**	**7.6 ***	**61.1**	**±**	**8.9 ***	45.2	±	7.0	44.9	±	7.9
*Prothrombine Time, %*	70–100	88.8	±	2.2	84.6	±	1.9	86.1	±	2.4	81.2	±	2.1	81.2	±	2.5	83.8	±	1.8	77.9	±	3.0	79.8	±	2.0
*APTT, sec*	26.3–40.3	33.2	±	0.9	34.2	±	0.7	36.4	±	2.6	39.0	±	4.0	32.9	±	1.0	32.2	±	0.8	37.0	±	3.4	32.8	±	1.2
*D-dimer, µg/mL*	≤0.50	2.1	±	0.4	1.9	±	0.2	4.2	±	2.0	4.1	±	1.4	**7.7**	**±**	**4.7 ***	**2.3**	**±**	**0.4 ***	2.0	±	0.2	2.5	±	0.4
*Ferritin (ng/L)*	30–400	1411.9	±	150.1	1418.4	±	150.3	1519.3	±	196.2	1803.1	±	201.0	1222.7	±	223.1	1377.3	±	175.1	1015.6	±	198.8	1141.6	±	153.6

Abbreviations: RBC = Red blood cell count; WBC = White blood cell count; PLT = Platelets; MCV = Mean corpuscular volume; MCH = Mean corpuscular hemoglobin; MCHC = Mean corpuscular hemoglobin concentration; RDW = Red Cell Distribution Width; NLR = Neutrophil–Lymphocyte Ratio; PLR = platelet–lymphocyte ratio; CRP = C reactive protein; APTT = Activated partial thromboplastin time; PT = Prothrombin time. (*) in bold represents significant differences (*p* < 0.05) using Student’s t test between the population with and without obesity.

**Table 3 jcm-12-03392-t003:** Pearson’s correlation between laboratory parameters and obesity of Chilean patients hospitalized with COVID-19.

Parameter	Day 1		Day 3		Day 7		Day 15	
	r	*p-*Value	r	*p-*Value	r	*p-*Value	r	*p-*Value
*Hematocrit, %*	0.062	0.384	0.059	0.441	−0.007	0.935	0.086	0.434
*Hemoglobin, gr/dL*	0.057	0.423	0.039	0.607	−0.040	0.634	0.065	0.557
*RBC, ×10^6^ μL*	0.053	0.451	0.056	0.463	0.047	0.571	0.101	0.361
*WBC, × 10^3^ μL*	0.146	**0.038 ***	0.155	**0.042 ***	0.199	**0.016 ***	0.104	0.348
*PLT, ×10^3^ μL*	0.059	0.405	0.007	0.922	0.012	0.890	−0.068	0.541
*MCV, fL*	−0.046	0.518	−0.036	0.634	0.007	0.937	−0.036	0.747
*MCH, pg*	−0.063	0.374	−0.068	0.374	−0.137	0.100	−0.082	0.456
*MCHC, gr/dL*	−0.078	0.268	−0.093	0.222	−0.168	**0.043 ***	−0.089	0.422
*RDW, %*	0.048	0.499	−0.07	0.362	−0.073	0.380	−0.090	0.418
*Lymphocytes,* *× 10^3^* *μL*	−0.021	0.771	0.082	0.284	−0.161	0.052	−0.021	0.847
*Monocytes, × 10^3^ μL*	0.111	0.117	0.081	0.288	0.075	0.367	−0.026	0.814
*Neutrophils, × 10^3^ μL*	0.151	**0.032 ***	0.133	0.081	0.219	**0.008 ***	0.115	0.297
*Eosinophils, × 10^3^ μL*	0.081	0.251	−0.050	0.514	0.014	0.867	0.118	0.286
*Basophils, × 10^3^ μL*	−	−	−0.057	0.459	0.073	0.379	0.076	0.492
*Prothrombine Time, %*	0.106	0.173	0.134	0.126	−0.082	0.379	−0.070	0.551
*NLR*	0.081	0.254	0.076	0.318	0.167	**0.045 ***	0.160	0.147
*PLR*	0.066	0.353	0.051	0.504	0.153	0.064	0.045	0.686
*CRP (gmg/L)*	0.099	0.180	−0.047	0.554	−0.063	0.459	0.006	0.958
*APTT, sec*	−0.072	0.358	−0.045	0.615	0.052	0.581	0.147	0.213
*D-dimer, ug/mL*	0.044	0.618	0.005	0.952	0.129	0.181	−0.130	0.328
*Ferritin (ng/L)*	−0.003	0.976	−0.086	0.337	−0.046	0.627	−0.032	0.806

(*) indicates significance for pairwise correlations (*p* < 0.05).

**Table 4 jcm-12-03392-t004:** Clinical results of patients with obesity hospitalized for COVID-19.

	With Obesity	Without Obesity	*p-*Value
*Hospital length of stay, days X + DS*	27.32 + 19.09	22.56 + 22.62	0.011
*ICU admission, n (%) **	49 (69.0)	60 (45.8)	0.001
*Invasive mechanical ventilation (n, %) ***	42 (59.2)	43 (32.8)	<0.001
*Death, n (%)*	11 (15.5)	21 (16.0)	0.920

* OR = 2.636 (IC 1.433–4.848; *p* = 0.0016), ** OR = 2.964 (IC 1.631–5.388; *p* = 0.0003).

## Data Availability

Not applicable.

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
