# Peer review of "Obesity Is Associated with Changes in Laboratory Biomarkers in Chilean Patients Hospitalized with COVID-19"

_jcm, 2023, doi:10.3390/jcm12103392_

Round 1
Reviewer 1 Report
Viscardi S and colleagues reported that obesity is associated with changes in laboratory biomarkers in Chilean patients hospitalized for COVID-19. The manuscript is interesting, but some comments should be addressed before its publication.
-There are some grammatical errors and typos in the manuscript.
-Please add the references in only format.
Abstract
-In this section the authors should indicate in the results section, the most important results and laboratory biomarkers with statistical significance.
Introduction
-In the next phrase: “As of January 2023, 670.191.206 laboratory confirmed human cases of COVID-19 have been notified to WHO with 6.823.120 deaths, …… of SARSCoV (2-4).” It is important to add a reference related with cases of COVID-19 (January 2023, e.g., WHO).
-Please define ACE2, Ang II, and RAS.
-In the next phrase: “although the majority of individuals eventually recover from infection, even those needing long hospitalization (9, 11-13).” The reference 12 is not appropriate for the information considered.
-In the following phrase: “In this context, it is important to note that obesity is characterized by adipose tissue renovation, and pro-inflammatory alteration of the adipokine profile (20).” The reference 12 is not correct for the information considered. I recommend a good reference for this paragraph (PMCID: PMC8583943).
Materials and Methods
-Please add how was laboratory biomarkers determined? In serum or plasma. The authors need to add the time of blood collection, which is the time when COVID-19 occurred?
-Why only 71 with obesity compared to 131 without obesity, I think the groups must be uniform.
-The authors should indicate the obesity grade according to BMI, were all patients in single degree of obesity? Which BMI grade?
Results
-How many patients with SARS-CoV-2 died? And how many patients with SARS-CoV-2 were on ventilation support? Please add these data in this section.
-Which could be the explanation about non-obese patients have more Mellitus diabetes 2 and ECV compared to obese patients (please discuss in discussion section).
-In the following phrase: “On evaluating comorbidities, there is a significant difference for chronic respiratory pathologies, where a greater number of patients with obesity present chronic respiratory disease (27.7% of patients), compared to 40.6% of not obese patients (p=0.001).” it is difficult to understand because in table 1 there other percentage data.
Discussion
-In the next phrase: “Our results suggest that patients with obesity have higher levels of lymphocytes on day 3, in contrast with the findings reported by Bobcakova et al. (30).” Please be clear and indicate that the study were made in non-obese patients (reference 30).
-The authors should include in this section, the data of DM2, ECV, Cough, Dyspnea, and CRP. Is there relationship between them in non-obese patients?
-Please add a phrase related with limitations of this study.
Conclusion
In this section, authors should specifically add data from groups analyzed with statistical significance; because de conclusion is very general.
Author Response
Viscardi S and colleagues reported that obesity is associated with changes in laboratory biomarkers in Chilean patients hospitalized for COVID-19. The manuscript is interesting, but some comments should be addressed before its publication.
A: Dear reviewer, thanks for your kind revision and suggestions, the manuscript is now improved
- There are some grammatical errors and typos in the manuscript.
A: corrected
- Please add the references in only format.
A: improved
ABSTRACT
- In this section the authors should indicate in the results section, the most important results and laboratory biomarkers with statistical significance. A: improved
INTRODUCTION
- In the next phrase: “As of January 2023, 670.191.206 laboratory confirmed human cases of COVID-19 have been notified to WHO with 6.823.120 deaths, …… of SARSCoV (2-4).” It is important to add a reference related with cases of COVID-19 (January 2023, e.g., WHO).
A: improved
- Please define ACE2, Ang II, and RAS. A: improved. Angiotensin converting enzyme 2 (ACE 2), angiotensin II (Ang II), renin-angiotensin system (RAS)
- In the next phrase: “although the majority of individuals eventually recover from infection, even those needing long hospitalization (9, 11-13).” The reference 12 is not appropriate for the information considered. A: corrected
- In the following phrase: “In this context, it is important to note that obesity is characterized by adipose tissue renovation, and pro-inflammatory alteration of the adipokine profile (20).” The reference 12 is not correct for the information considered. I recommend a good reference for this paragraph (PMCID: PMC8583943).
A: We appreciate the reviewer's recommendation. The recommended article was considered.
MATERIALS AND METHODS
- Please add how was laboratory biomarkers determined? In serum or plasma. The authors need to add the time of blood collection, which is the time when COVID-19 occurred?
A: Data collection (page 3) details how biomarkers were determined (sample collection at the time of hospital admission), laboratory methodologies used to measure each of the laboratory parameters). In addition, in numeral 2.1. Study design and participants details hospital admission period.
- Why only 71 with obesity compared to 131 without obesity, I think the groups must be uniform.
A: The study considers patients hospitalized for a period of time, noting that 35% of the patients included in the study presented obesity. Although ideally the groups should have a similar number of subjects, it is possible to consider a larger number of subjects who do not present the condition under study, in this case obesity. We believe that limiting the number of non-obese patients would not modify the observed results, considering that there are no significant differences by sex and age between both groups of patients (obese and non-obese).
- The authors should indicate the obesity grade according to BMI, were all patients in single degree of obesity? Which BMI grade?
A: Obesity or normal weight status was obtained from the medical history record of each patient, which does not detail the BMI value in all cases since it was performed by subjective global assessment. Thus, the subjects included in the study were grouped as obese and non-obese. As this is a limitation of the study, it is made explicit in the discussion.
RESULTS
- How many patients with SARS-CoV-2 died? And how many patients with SARS-CoV-2 were on ventilation support? Please add these data in this section.
A: This requested information is presented in the results text (page 4). In addition, based on the reviewer's suggestion, the association of obesity with clinical outcomes such as length of hospital stay, admission to intensive care unit, Invasive Mechanical ventilation (IMV) and death was further evaluated. This information is included in the results text (3.4 Clinical Outcomes) and Table 4 (page 10). These results are discussed later (page 11).
- Which could be the explanation about non-obese patients have more Mellitus diabetes 2 and ECV compared to obese patients (please discuss in discussion section).
A: Discussion improved in the manuscript
- In the following phrase: “On evaluating comorbidities, there is a significant difference for chronic respiratory pathologies, where a greater number of patients with obesity present chronic respiratory disease (27.7% of patients), compared to 40.6% of not obese patients (p=0.001).” it is difficult to understand because in table 1 there other percentage data.
A: The text is corrected according to what is presented in table 1.
DISCUSSION
- In the next phrase: “Our results suggest that patients with obesity have higher levels of lymphocytes on day 3, in contrast with the findings reported by Bobcakova et al. (30).” Please be clear and indicate that the study were made in non-obese patients (reference 30). A: improved
- The authors should include in this section, the data of DM2, ECV, Cough, Dyspnea, and CRP. Is there relationship between them in non-obese patients?
A: The analysis of these results is included in the discussion.
- Please add a phrase related with limitations of this study. A: A paragraph is included regarding the limitations of the study.
Conclusion
- In this section, authors should specifically add data from groups analyzed with statistical significance; because de conclusion is very general. A: improved

Reviewer 2 Report
The authors examined the laboratory biomarkers between obesity and non-obesity Chilean patients hospitalized for covid-19. Now it is common sense that obesity and diabetes are the risk factor the severity and mortality in patients with covid-19 infection. The authors found that obesity patients exhibited higher incidence for chronic respiratory pathology, anosmia and dysgeusia. Their elevated laboratory biomarker such as inflammatory parameters and D-dimer were observed compared to non-obese patients.
1 1. In results 3.1 and second paragraph of discussion, the authors stated that a greater number of patients with obesity presented chronic respiratory disease (27.7% of patients) compared to 40.6% of not obese patients. However, it is not consistent with the data showed in the table 1.
2 2. Table 3 is exactly same with Table 2, except title.
3 3. The number of patients for covid-19 is small when considering its global pandemic. And the number of non-obesity patients is almost double than the number of obesity patients.
Author Response
Comments and Suggestions for Authors
REVIEWER 2
The authors examined the laboratory biomarkers between obesity and non-obesity Chilean patients hospitalized for covid-19. Now it is common sense that obesity and diabetes are the risk factors the severity and mortality in patients with covid-19 infection. The authors found that obesity patients exhibited higher incidence for chronic respiratory pathology, anosmia and dysgeusia. Their elevated laboratory biomarker such as inflammatory parameters and D-dimer were observed compared to non-obese patients.
A: Dear reviewer, thanks for your kind revision and suggestions, the manuscript is now improved
- In results 3.1 and second paragraph of discussion, the authors stated that a greater number of patients with obesity presented chronic respiratory disease (27.7% of patients) compared to 40.6% of not obese patients. However, it is not consistent with the data showed in the table 1. A: It was corrected in the discussion text.
- Table 3 is exactly same with Table 2, except title.
A: Corrected. Table 3 presents correlation values and statistical significance.
- The number of patients for covid-19 is small when considering its global pandemic. And the number of non-obesity patients is almost double than the number of obesity patients.
A: This study was conducted in a single clinical center in southern Chile, which cares for part of the region's population, which explains the limited number of patients included in the study. In addition, the region where the study was carried out presents an important component of the native population and multidimensional poverty. Therefore, a paragraph was included in the discussion where the small number of patients is considered within the limitations of the study.

Reviewer 3 Report
This is an interesting topic which also increases awareness as to the evolution of COVID-19 in different groups an their biomarkers. Obesity is unquestionably a risk factor that has been interesting for many investigators during the pandemic and has resulted controversial in many aspects. It is difficult to assess all the factors that are relevant to a biomarker during a disease, but some information is important in order to evaluate the data presented by the authors.
Considering these controversies I would suggest some modifications and texts regarding the patient selection and evaluation.
a) Morbid obesity is a term not longer suggested, but severe obesity or Grade III according to the World Health Classification in 2019. Despite "morbid"is a word that is still commonly used in medical literature, current texts suggest changing these words as it can be perceived as offensive and the objective of literature and medical practice is to encourage the patient to take action in the problem. The 2019 classification by the World Health organization includes the term Super obesity for patients with a BMI of 50 kg/m2 or more.
References:
1) Class III Obesity (Formerly Known as Morbid Obesity). The Cleveland Clinic. Access: https://my.clevelandclinic.org/health/diseases/21989-class-iii-obesity-formerly-known-as-morbid-obesity#:~:text=Now%2C%20healthcare%20providers%2C%20researchers%20and,place%20of%20%E2%80%9Cmorbid%20obesity.%E2%80%9D
2) Volger S, Vetter ML, Dougherty M, Panigrahi E, Egner R, Webb V, Thomas JG, Sarwer DB, Wadden TA. Patients' preferred terms for describing their excess weight: discussing obesity in clinical practice. Obesity (Silver Spring). 2012 Jan;20(1):147-50. doi: 10.1038/oby.2011.217. Epub 2011 Jul 14. PMID: 21760637; PMCID: PMC3310899.
b) The group without obesity needs to be better characterized. Their mean or median BMI should be included and the percentage of underweight patients (if any) should also be reported. Rapid or recetn weight loss, as well as malnutrition have also been described as risk factors for severity in normal or low weight patients.
c) Authors should describe any additional elimination criteria. For example, if they included (or not) patients with malignant neoplasms, chronic steroid use or previous bariatric surgery. All of these comorbidities affect weight and biomarkers. If data was nor available for this information, they should disclose it.
d) The time that patients presented with symptoms before hospitalization is relevant to the study as well.
e) some biomakers in the text are also increased when a bacterial infection is added to COVID-19, which may also account for the severity at baseline or worsening symptoms during hospital stay. the authors should also describe the percetage of patients that had a bacterial coinfection or at least required antibiotics. The biomarkers also change with nutritional status, even during inflammation, which makes it necessary to know if the change during the follow up is a reflection of an increased inflammation or only a reflection of the nutritional reserves of the patients.
References
1) Mason CY, Kanitkar T, Richardson CJ, Lanzman M, Stone Z, Mahungu T, Mack D, Wey EQ, Lamb L, Balakrishnan I, Pollara G. Exclusion of bacterial co-infection in COVID-19 using baseline inflammatory markers and their response to antibiotics. J Antimicrob Chemother. 2021 Apr 13;76(5):1323-1331. doi: 10.1093/jac/dkaa563. PMID: 33463683; PMCID: PMC7928909.
2) Namaste SM, Rohner F, Huang J, Bhushan NL, Flores-Ayala R, Kupka R, Mei Z, Rawat R, Williams AM, Raiten DJ, Northrop-Clewes CA, Suchdev PS. Adjusting ferritin concentrations for inflammation: Biomarkers Reflecting Inflammation and Nutritional Determinants of Anemia (BRINDA) project. Am J Clin Nutr. 2017 Jul;106(Suppl 1):359S-371S. doi: 10.3945/ajcn.116.141762. Epub 2017 Jun 14. PMID: 28615259; PMCID: PMC5490647.
e) The text is not clear as to whether the patients were hospitalized due to respiratory distress or if only the patients that required ventilatory support were included. If the patients included also cases with only oxygen support, the percentage of patients that required intubation is necessary in both groups. The changes in biomarkers were predictive of intubation at some point?
f) It is also not clear if the evaluations were performed only in patients that were hospitalized for the complete 15 days or the number of patients evaluated at each point. It is clear that some patients may have shorter stays than others, but it may be interesting to know if there are also differences between the patients that had shorter stays than longer hospitalizations or Intensive Care stay.
g) I would need to know if the percentage of patients hospitalized for COVID-19 with obesity are higher that the percentage of patients with obesity in the general population for Chile. According to some data, obesity rates have increased in this country in the last years. From the numbers, it seems that the percentage of obesity in the whole group is 35.1%, maybe similar to the general population, and higher in women than in men.
Author Response
Comments and Suggestions for Authors
REVIEWER 3
This is an interesting topic which also increases awareness as to the evolution of COVID-19 in different groups an their biomarkers. Obesity is unquestionably a risk factor that has been interesting for many investigators during the pandemic and has resulted controversial in many aspects. It is difficult to assess all the factors that are relevant to a biomarker during a disease, but some information is important in order to evaluate the data presented by the authors.
Considering these controversies I would suggest some modifications and texts regarding the patient selection and evaluation.
A: Dear reviewer, thanks for your kind revision and suggestions, the manuscript is now improved
- a) Morbid obesity is a term not longer suggested, but severe obesity or Grade III according to the World Health Classification in 2019. Despite "morbid"is a word that is still commonly used in medical literature, current texts suggest changing these words as it can be perceived as offensive and the objective of literature and medical practice is to encourage the patient to take action in the problem. The 2019 classification by the World Health organization includes the term Super obesity for patients with a BMI of 50 kg/m2 or more.A: corrected
- b) The group without obesity needs to be better characterized. Their mean or median BMI should be included and the percentage of underweight patients (if any) should also be reported. Rapid or recetn weight loss, as well as malnutrition have also been described as risk factors for severity in normal or low weight patients.
A: Obesity or normal weight status was obtained from the medical history record of each patient, which does not detail the BMI value in all cases, since in hospital clinical practice subjective global assessment is performed. Thus, the subjects included in the study were grouped as with or without obesity. Low weight patients were excluded from the study, which is detailed in materials and methods (section 2.1). One of the limitations of the study is related to the information in the patient's medical history, which may have some incomplete records, which is explained in the discussion.
- c) Authors should describe any additional elimination criteria. For example, if they included (or not) patients with malignant neoplasms, chronic steroid use or previous bariatric surgery. All of these comorbidities affect weight and biomarkers. If data was nor available for this information, they should disclose it.
A: In materials and methods, exclusion criteria are detailed (section 2.1). In addition, a paragraph is included in the discussion regarding the limitations of the study, among which some incomplete records of the medical history are considered, for which it was not possible to record chronic use of steroids and previous bariatric surgery.
- d) The time that patients presented with symptoms before hospitalization is relevant to the study as well.
A: Based on the reviewer's suggestion, this antecedent is added to the results text (page 4). The results show that no significant differences were observed regarding symptoms prior to hospitalization (5.43 + 3.8 days obese patients vs 4.73 + 5.1 days in not obese patients, p=0.162).
- e) some biomakers in the text are also increased when a bacterial infection is added to COVID-19, which may also account for the severity at baseline or worsening symptoms during hospital stay. the authors should also describe the percetage of patients that had a bacterial coinfection or at least required antibiotics. The biomarkers also change with nutritional status, even during inflammation, which makes it necessary to know if the change during the follow up is a reflection of an increased inflammation or only a reflection of the nutritional reserves of the patients.
A: The 4.2% of patients with obesity and 3.8% of patients without obesity presented coinfection during hospitalization, less than that described in a recent meta-analysis. It is included in the discussion of results of inflammatory parameters (CPR and ferritin)
- e) The text is not clear as to whether the patients were hospitalized due to respiratory distress or if only the patients that required ventilatory support were included. If the patients included also cases with only oxygen support, the percentage of patients that required intubation is necessary in both groups. The changes in biomarkers were predictive of intubation at some point?
A: The patients were hospitalized due to respiratory distress and not all required invasive mechanical ventilation. This information is presented in the results text (page 4). In addition, based on the reviewer's suggestion, the association of obesity with clinical outcomes such as length of hospital stay, admission to intensive care unit, Invasive Mechanical ventilation (IMV) and death was further evaluated. This information is included in the results text (3.4 Clinical Outcomes) and Table 4 (page 10). These results are discussed later (page 11).
- f) It is also not clear if the evaluations were performed only in patients that were hospitalized for the complete 15 daysor the number of patients evaluated at each point. It is clear that some patients may have shorter stays than others, but it may be interesting to know if there are also differences between the patients that had shorter stays than longer hospitalizations or Intensive Care stay.
A: Section 3.4 and Table 4 present the association of obesity with clinical outcomes, including length of hospital stay. Thus, length of hospital stay (27.32 + 19.09 days in obese vs 22.56 + 22.62 days not obese, p=0.011), which is also analyzed in the discussion.
- g) I would need to know if the percentage of patients hospitalized for COVID-19 with obesity are higher that the percentage of patients with obesity in the general population for Chile. According to some data, obesity rates have increased in this country in the last years. From the numbers, it seems that the percentage of obesity in the whole group is 35.1%, maybe similar to the general population, and higher in women than in men.
A:The Chilean National Health Survey corresponds to a cross-sectional population survey of a national nature that is carried out periodically in the country. The last study showed that 31.2% of the general Chilean population had grade I and II obesity (28.6% men vs 33.7% women), while 3.2% had grade III obesity (1.7% men vs 4.7 women). . In addition, the Chilean National Health Survey shows that obesity affects 35.5% of the general population of the La Araucanía region, where this study was conducted.
Based on the above, the results of this study show that the percentage of obese patients hospitalized for COVID-19 is slightly higher than that described at the national level for the general population and similar to that described for the region where the research was conducted. being higher in women than in men.
Reference
Ministerio de Salud. Informe Encuesta Nacional de Salud 2016-2017: Estado Nutricional. Santiago de Chile; 2018, 42p. Avaible at: https://goo.gl/oe2iVt

Round 2
Reviewer 1 Report
I have no comments.
Author Response
Thanks

Reviewer 2 Report
The manuscript was greatly improved by the authors. However, the authors should improve the presentation quality of table 1 and 3 and correct their data in table 4.
Author Response
Dear Reviewer,
Thank you for your comments, we make the pertinent modifications
Best Regards
